# The role of sex education in withdrawal use: Prevalence and correlates among a nationally representative sample of adolescents and young adults

John L. Ferrand[1,2], Arthur H. Owora[3], Alexandra T. Hughes-Wegner[1],
Eric R. Walsh-Buhi[1,4,5]*

1 Department of Applied Health Science, Indiana University School of Public Health-Bloomington, Bloomington, Indiana, United States of America, 2 Leidos, San Diego, California, United States of America, 3 Department of Pediatrics, Indiana University School of Medicine, Indianapolis, Indiana, United States of America, 4 Center for Sexual Health Promotion, Indiana University, Bloomington, Indiana, United States of America, 5 The Kinsey Institute, Indiana University, Bloomington, Indiana, United States of America

* erwals@iu.edu

## Abstract

### Objectives

Instances of withdrawal, the practice of removing a penis from a vagina before ejaculation to prevent pregnancy, have increased in some US populations over the past two decades. There is a paucity of research, however, on the prevalence and correlates of withdrawal among adolescents and young adults (AYAs). This study sought to determine the prevalence and correlates of withdrawal use in a representative sample of AYAs in the US, highlighting the association between receipt of informal sexuality education and withdrawal use.

### Study design

We analyzed cross-sectional National Survey of Family Growth data from AYAs surveyed between 2011 and 2019 (15–24 years; N = 14,262). Prevalence of withdrawal was determined using different sexual activity time-points (at first sex and ever) and reference periods (past 3 and 12 months). Logistic regression models were used to identify correlates of withdrawal alone and combined with at least one other method.

### Results

Across all waves and reference periods, the prevalence of withdrawal was higher among AYAs who combined it with at least one other method (ever [overall]=15.15, SE = 0.58) compared with those who used it as the only method (ever [overall]=8.32, SE = 0.38). Compared to the 2011–2013 wave, those in the 2015–2017 wave had

**Data availability statement:** The data underlying the results presented in the study are available from U.S. Department of Health and Human Services (https://www.cdc.gov/nchs/nsfg/index.htm).

**Funding:** The author(s) received no specific funding for this work.

**Competing interests:** The authors have declared that no competing interests exist.

greater odds of reporting using only withdrawal in the past 3 months (AOR: 1.50; 95% CI: 1.02, 2.21). The same trend was seen in those who used withdrawal with another method at last sex in the past 3 and 12 months. Withdrawal alone or combined with another method varied by receipt of informal sexuality education topics (e.g., methods of birth control vs STIs).

## Conclusion

Variations in use of withdrawal based on type of informal sexuality education received suggests that different motivations might inform interventions in promoting effective reproductive health practices.

---

## Introduction

Sexually transmitted infections (STIs) and human immunodeficiency virus (HIV) disproportionally impact adolescents and young adults (AYA) in the United States (US). While AYAs are often defined by various developmental, relational, and socioemotional milestones, they are broadly defined as being between the ages of 10–25 [1–4]. Persons aged 15–24 years, for example, represented more than half (51%) of reported STI diagnoses in 2021 and those aged 13–24 years represented just under one-fifth (19%) of reported HIV diagnoses in 2021 [5,6]. Additionally, instances of unintended pregnancy among AYAs in the US, while declining overall during the past two decades, are still high with 75% of pregnancies reported by young women aged 15–19 classified as unintended in 2011 [7]. A recent study found that, even though the overall rate of unwanted pregnancies decreased between 2009 and 2015, the rate did not decrease for women aged 25 years or older [8]. Efforts aimed at reducing instances of STI/HIV and unintended pregnancy generally focus on similar protective behaviors such as condom use or abstinence [9–11], but this approach may ignore nuanced differences between how one protects oneself from an STI/HIV compared to unintended pregnancy [12,13].

Withdrawal, the practice of removing a penis from a vagina before ejaculation to prevent pregnancy, is considerably less effective at protecting against both STI/HIV and unintended pregnancy [14] compared to other methods. While all forms of birth control require correct and consistent use to maximize their efficacy [15,16], estimates of successful withdrawal use vary [17–19], with approximately 60% of Americans aged 15–44 years reportedly using withdrawal between 2006 and 2010 [20]. More recent estimates from the National Survey of Family Growth (NSFG) suggests the percentage of women aged 15–19 years who have ever used withdrawal has been steadily increasing since 2002 (55%), with 60% reporting ever using withdrawal between 2011–2015 [21]. An updated frequency and trend analysis of withdrawal use is critical, as estimates suggest that the probability of experiencing withdrawal failure is nearly 1 in 5 (18%) [22], increasing risk for an unintended pregnancy.

While many AYAs receive some kind of formal sex education throughout their school system in the US, it is often abstinence-based and does not discuss the different types of contraception, including withdrawal [23]. Participation in sexuality education programs

or interventions in adolescence has been associated with improved sexual health outcomes in later life [24–26]. However, formal sexuality education, often delivered by instructors in a school, church, youth center, or other community-based setting [27], has declined in recent years. Given the knowledge gap and the lack of efficacy in abstinence-based education in sexual and reproductive health (SRH) promotion [28,29], it is vital to understand how young people are being educated about their sexual health. While formal sex education–sexuality education delivered through a formal institution such as a school (e.g., health class, after school programming), church, or community center–across the US has been lacking, a majority of young people receive sex education through informal means such as the Internet, peers, and family [30]. Despite the Internet being one of the most prominent informal means [31], family members and other caretakers play a vital role in teaching AYAs about SRH topics [32–36]. Nevertheless, little is known about how discussing SRH topics with a parent or caregiver influences contraceptive use. A growing body of research focuses on informal means of sexuality education and its association with contraceptive use, associating it with other SRH outcomes, such as beliefs about sex [33] or access to information about sex [37].

The purpose of this paper is to provide an update on the prevalence and correlates of withdrawal as a contraceptive method amongst a representative sample of AYAs in the United States (US), focusing on the association between receipt of informal sexuality education and one's withdrawal use.

## Materials and methods

### Data source

Four waves (2011–2013, 2013–2015, 2015–2017, and 2017–2019) of the NSFG, a cross-sectional, household survey were examined. The NSFG collects data using computer-assisted personal interviews of men and women aged 15–49. The NSFG utilizes a stratified, multistage, clustered, probability sampling design to report nationally representative estimates, meaning that the survey is designed to be nationally representative of men and women aged 15–49 living in households in the United States. These data are de-identified and publicly available. Additional information on the data collection procedures, responses rates, sample design, variance estimation, and fieldwork procedures for the most recent NSFG surveys have been published previously [38].

### Study sample

The data file for this study contains nine years of interviews spanning from 2011 to 2019. These four combined datasets contain a total of 42,062 interviews including 14,262 adolescents and young adults, aligning with our defined target population of AYAs between the age of 10–25. Our total unweighted analytic sample included 7,410 non-pregnant females and 6,852 males aged 15–24 years interviewed between 2011 and 2019. Institutional Review Board approval was not sought for this secondary data analysis.

### Outcomes

**Withdrawal.** Our analysis examined the prevalence and correlates of withdrawal use alone and in combination with at least one other contraceptive method separately. Four reference (recall) periods were used to define withdrawal use: 1) ever use, 2) at first sex, 3) past 12 months, and 4) past 3 months [39].

**Informal sexuality education.** Respondents were asked on the NSFG whether they had discussed any of the following topics with a parent or caregiver: how to say no to sex; birth control methods; where to get birth control; how to use a condom; STIs; preventing HIV/AIDS; and waiting until marriage to have sex. A dichotomous indicator variable was created for each topic.

### Covariate measures

**Age.** We restricted the sample to only respondents who were 15–24 years old and adjusted for age as a potential confounding variable in our final analysis models.

**Sex.** Sex of the respondent was coded as either female or male.

**Race/ethnicity.** Similar to previous studies [40,41] we coded race/ethnicity into four categories: non-Hispanic white; non-Hispanic black; Hispanic; and non-Hispanic other or multiple races.

**Education.** AYAs were classified as belonging to one of six education levels: 9th grade or less, some high school, GED/high school diploma, some college, college degree, or graduate/professional degree.

**Religious affiliation.** Respondents selected from 11 response categories (e.g., none, Catholic, Jewish, Baptist, Methodist/African Methodist) which was coded into a dichotomous indicator (currently affiliated with vs. not currently affiliated with any religion).

**Formal sexuality education.** Respondents were asked whether they had received specific instruction on sexuality education topics (the same as in the informal sexuality item) before they were 18 years of age in a formal setting such as a school, church, or community center. A dichotomous indicator variable was created for each topic ("Yes" or "No").

## Statistical analysis

Survey weighted descriptive statistics were used to summarize responses of study participants across the four NSFG data waves. The prevalence of withdrawal was summarized by NSFG survey wave, receipt of informal sexuality education, and participant characteristics.

Multiple binary logistic regression models were employed to identify correlates of withdrawal use. This method was selected due to its combined simplicity and effectiveness in modeling the odds of dichotomous event (in this case using withdrawal or not using withdrawal) while accounting for mixed independent predictor variables. All statistical analyses were implemented using SAS 9.4 [42]. Appropriate SAS procedures were used to take into account the complex sampling design of the NSFG and to produce unbiased estimates representative of the U.S. AYA population. Statistical significance was assessed at an alpha of 0.05.

## Results

Table 1 summarizes the sociodemographic characteristics of the study sample. Briefly, the mean age was 19.59 years (SE = 0.04), 50.81% were male, 54.07% were non-Hispanic Whites, 26.45% had at least some college education, 88.93% were not cohabiting with an opposite sex partner, 72.00% had a religious affiliation, and 97.15% reported receipt of some formal sexuality education in the past.

### Withdrawal use as the only contraceptive method

The prevalence of ever use of withdrawal as the only contraceptive method increased from 7.77% in 2011–2013 and 2013–2015 to 9.30% in 2015–2017 and declined to 8.51% in 2017–2019 (Fig 1). Except for withdrawal use at first sex, a similar trend was observed for other reference periods (past 12- and 3-months). Relative to 2011–2013, there was a decline and subsequent increase in the prevalence of withdrawal use for first sex during survey waves 2013–2015 and 2015–2017, respectively. The prevalence of withdrawal use across the different survey waves and reference periods was generally higher for respondents who were male and Hispanic, had at least some college education, no current religious affiliation, cohabiting with a partner, and those who received some formal sexuality education. Table 2 provides additional detail on the prevalence of withdrawal across sociodemographic characteristics and survey wave.

Receipt of STI-related informal sexuality education was associated with lower odds of ever using withdrawal (AOR: 0.76; 95% CI: 0.60, 0.96; Table 3) and lower odds of using withdrawal in the past 12 months (AOR: 0.72; 95% CI: 0.55, 0.94). AYAs who completed the NSFG between 2017 and 2019 had greater odds of reporting withdrawal use in the past 3 months compared to those who completed the NSFG between 2011 and 2013.

**Table 1.  Unweighted and weighted distributions of sociodemographic characteristics among adolescents and young adults aged 15-24 years in NSFG between 2011 and 2019 (N = 14,262).**

| Characteristic | Unweighted N | Weighted % (SE) |
|---|---|---|
| Overall | 14,262 | 100% |
| Survey Wave | | |
| 2011-2013 | 3895 | 25.24 (1.10) |
| 2013-2015 | 3693 | 25.25 (1.09) |
| 2015-2017 | 3194 | 24.93 (1.32) |
| 2017-2019 | 3480 | 24.57 (1.28) |
| Age (M, SE) | 19.59 (0.04) | |
| Sex | | |
| Female | 7410 | 49.19 (0.70) |
| Male | 6852 | 50.81 (0.70) |
| Race/Ethnicity | | |
| Hispanic | 4042 | 22.83 (1.09) |
| Non-Hispanic white | 6254 | 54.07 (1.16) |
| Non-Hispanic black | 3007 | 15.32 (0.75) |
| Non-Hispanic other | 959 | 7.78 (0.75) |
| Education level | | |
| 9th grade or less | 2323 | 14.13 (0.45) |
| Some high school | 3802 | 23.22 (0.60) |
| GED/high school diploma | 3828 | 25.53 (0.65) |
| Some college | 3085 | 26.45 (0.89) |
| College degree | 1172 | 10.10 (0.45) |
| Graduate or professional degree | 52 | 0.57 (0.13) |
| Has a religious affiliation | | |
| No | 3797 | 28.00 (0.79) |
| Yes | 10,465 | 72.00 (0.79) |
| Currently cohabitating with a partner | | |
| No | 12,976 | 88.93 (0.46) |
| Yes | 1286 | 11.07 (0.46) |
| Received any formal sexuality education | | |
| No | 440 | 2.85 (0.22) |
| Yes | 13,822 | 97.15 (0.22) |

## Withdrawal use combined with other contraceptive methods

In contrast to the use of withdrawal as the sole method of contraception, the prevalence of withdrawal use in combination with other contraceptive methods was higher but varied in similar patterns across survey waves (Fig 2), reference periods, and respondents' socio-demographic factors. Tables 4 and 5 depict these patterns.

Similar to those reporting withdrawal use as the only contraceptive method, receipt of STI-related informal sexuality education was associated with lower odds of ever using withdrawal (AOR: 0.80; 95% CI: 0.67, 0.96) and lower odds of using withdrawal in the past 12 months (AOR: 0.79; 95% CI: 0.65, 0.96). Moreover, receipt of informal sexuality education related to other methods of birth control was associated with higher odds of ever using withdrawal (AOR: 1.33; 95% CI: 1.09, 1.63) and higher odds of having used withdrawal in the past 12 months (AOR: 1.38; 95% CI: 1.11, 1.71) and past 3 months (AOR: 1.30; 95% CI: 1.01, 1.67). The odds of reporting withdrawal use in the past 12 months (AOR: 1.42; 95%

**Fig 1. Prevalence of withdrawal as the only contraceptive method across NSFG waves.**

CI: 1.04, 1.95) and in the past 3 months (AOR: 1.45; 95% CI: 1.05, 2.01) were greater for those in the 2015–2017 wave of the NSFG compared to the 2011–2013 wave.

## Discussion

The goal of the current study was to provide an updated estimate of the prevalence of withdrawal as a contraceptive method among a nationally representative sample of American AYAs and to examine the relationship with informal sexuality education using data from the 2019 NSFG data set. Our findings show that the prevalence of withdrawal varies depending on the context of its use, reference period, and survey wave and that discussing certain sexuality education topics with one's parents or caregivers is associated with one's likelihood of using withdrawal.

In general, the proportion of AYAs aged 15–24 who reported using withdrawal, either alone or with another method, was relatively small, suggesting that withdrawal is not commonly used. However, more respondents reported using it in combination with another method than using it alone. Our results suggest these differences may be influenced by informal sexuality education on STIs and birth control, depending on context and reference period. The 2015–2017 survey wave was occasionally associated with higher odds of withdrawal use compared to the 2011–2013 wave. Limited research exists on withdrawal use at first sex, and its low prevalence may be due to the lack of emphasis in formal sexuality education in US schools [23], potentially leading to underreporting due to social desirability bias [43]. We also cannot downplay the role that funding and government support play in influencing sexual health practices. The changes in withdrawal use at first sex in 2013–2015 could be the result of the Barrack Obama administration's efforts to reduce funding in abstinence only programs and increase funding for comprehensive sexuality education during his first and 2nd terms. However, the increase of withdrawal use during the 2015–2017 NSFG wave may be partially explained by early shifts in political discourse around sexual health [43–45]. The continuation of this trend could be due to the subsequent funding cuts to comprehensive sexual health programs during the Trump administration (2016–2020) [43–45].

The higher odds of withdrawal (combined with other contraceptive methods) associated with informal sexuality education covering topics such as STIs and birth control methods suggests that those exposed to discussions on these topics may be better equipped to seek out other forms of protection in the form of birth control and condoms that supplement withdrawal [46,47]. The lower odds of withdrawal use as the only contraceptive method among AYAs who received

**Table 2. Prevalence of withdrawal as the only contraceptive method in NSFG respondents aged 15-24 years between 2011 and 2019 (N=14,262).**

| Predictor Variable | Used withdrawal as the only method | | | |
| --- | --- | --- | --- | --- |
| | Ever | At first sex | In past 12 mos | In past 3 mos |
| | Weighted % (SE) | | | |
| Overall | 8.32 (0.38) | 2.29 (0.18) | 6.79 (0.34) | 5.99 (0.33) |
| Age (M, SE) | 20.87 (0.10) | 20.57 (0.19) | 20.95 (0.11) | 21.07 (0.11) |
| Sex | | | | |
| Female | 7.41 (0.53) | 2.06 (0.27) | 5.99 (0.47) | 5.28 (0.46) |
| Male | 9.20 (0.57) | 2.52 (0.26) | 7.57 (0.51) | 6.67 (0.49) |
| Race/Ethnicity | | | | |
| Hispanic | 90.68 (0.83) | 2.68 (0.38) | 7.58 (0.82) | 6.70 (0.82) |
| Non-Hispanic white | 7.69 (0.52) | 2.12 (0.25) | 6.18 (0.46) | 5.51 (0.44) |
| Non-Hispanic black | 9.02 (0.87) | 2.34 (0.38) | 7.90 (0.85) | 7.11 (0.82) |
| Non-Hispanic other | 8.41 (1.15) | 2.20 (0.69) | 6.52 (1.03) | 5.01 (1.04) |
| Education level | | | | |
| 9th grade or less | 2.35 (0.50) | 0.94 (0.40) | 1.61 (0.32) | 1.35(0.29) |
| Some high school | 5.86 (0.70) | 1.68 (0.31) | 4.59 (0.63) | 3.79 (0.53) |
| GED/high school diploma | 11.20 (0.81) | 3.01 (0.41) | 9.21 (0.73) | 8.12 (0.69) |
| Some college | 9.94 (0.71) | 2.94 (0.37) | 8.10 (0.65) | 7.30 (0.67) |
| College degree | 10.20 (1.24) | 1.91 (0.60) | 9.01 (1.18) | 8.08 (1.16) |
| Graduate or professional degree | 19.02 (8.17) | 4.80 (3.30) | 16.59 (8.11) | 16.59 (8.12) |
| Has a religious affiliation | | | | |
| No | 9.07 (0.78) | 1.90 (0.28) | 7.67 (0.72) | 6.96 (0.68) |
| Yes | 8.03 (0.45) | 2.44 (0.24) | 6.45 (0.39) | 5.61 (0.37) |
| Currently cohabitating with a partner | | | | |
| No | 7.95 (0.38) | 2.35 (0.19) | 6.43 (0.34) | 5.53 (0.33) |
| Yes | 11.30 (1.37) | 1.85 (0.58) | 9.73 (1.19) | 9.63 (1.19) |
| Received any formal sexuality education | | | | |
| No | 6.22 (1.51) | 2.12 (0.80) | 3.22 (0.96) | 3.81 (1.36) |
| Yes | 8.38 (0.38) | 2.30 (0.18) | 6.90 (0.34) | 6.05 (0.33) |

informal sexuality education on STIs may reflect a wariness in respondents' use of withdrawal when they are aware of possible negative sexual health outcomes such as STIs. These results are similar to findings from other studies examining receipt of similar topics in formal settings, suggesting that receipt of formal and informal sexuality education may be associated with similar contraceptive choices [25,40,48].

Previous studies of contraception have often focused on the relationship between formal sexuality education and contraceptive use (i.e., condoms or hormonal birth control) [40]. Yet, there is increasing evidence from recent studies noting withdrawal as a contraceptive method is increasing in the US. Interestingly, there may be a level of trust occurring between long-term or consistent partners influencing their decision to use withdrawal in lieu of a condom, despite the remaining risk of pregnancy and STI/HIV [49]. It is important to note, however, that this evidence has been mostly limited to female populations and recent reports of sexual intercourse (e.g., past 3 months) [50]. Our study adds to existing literature by examining the use of withdrawal as the only or complimentary method of contraception among both male and female AYAs using different recall or reference periods.

**Table 3. Binary logistic regression results of the association between topics of informal sexuality education and using withdrawal as the only method among adolescents and young adults aged 15-24 years in NSFG between 2011 and 2019 (N = 14,262).**

| Predictor Variable | Ever | At first sex | In past 12 mos | In past 3 mos |
|---|---|---|---|---|
| | AOR (95% CI) | AOR (95% CI) | AOR (95% CI) | AOR (95% CI) |
| Survey Wave[a] | | | | |
| 2013-2015 | 0.94 (0.66, 1.33) | 0.66 (0.37, 1.18) | 1.06 (0.73, 1.53) | 1.18 (0.80, 1.73) |
| 2015-2017 | 1.16 (0.84, 1.62) | 0.83 (0.47, 1.46) | 1.31 (0.92, 1.87) | **1.50 (1.02, 2.21)[c]** |
| 2017-2019 | 1.05 (0.77, 1.43) | 1.05 (0.62, 1.78) | 1.07 (0.77, 1.50) | 1.19 (0.84, 1.70) |
| Sexuality education topic discussed[b] | | | | |
| Any topics | 0.99 (0.70, 1.39) | 0.77 (0.41, 1.45) | 1.12 (0.78, 1.60) | 1.15 (0.78, 1.71) |
| Sexually transmitted infections | **0.76 (0.60, 0.96)[c]** | 1.14 (0.75, 1.73) | **0.72 (0.55, 0.94)[c]** | 0.79 (0.59, 1.05) |
| How to prevent HIV/AIDS | 0.86 (0.67, 1.11) | 0.84 (0.51, 1.39) | 0.90 (0.67, 1.20) | 0.80 (0.59, 1.09) |
| How to say no to sex | 1.05 (0.77, 1.43) | 1.04 (0.63, 1.72) | 1.01 (0.72, 1.40) | 1.13 (0.80, 1.61) |
| How to wait until marriage to have sex | 1.18 (0.91, 1.54) | 0.93 (0.59, 1.46) | 1.17 (0.87, 1.57) | 1.12 (0.82, 1.54) |
| How to use a condom | 0.84 (0.63, 1.12) | 0.86 (0.54, 1.37) | 0.83 (0.61, 1.13) | 0.83 (0.60, 1.16) |
| Methods of birth control | 1.12 (0.86, 1.47) | 1.12 (0.70, 1.80) | 1.11 (0.84, 1.48) | 1.07 (0.79, 1.45) |
| Where to get birth control | 1.14 (0.87, 1.51) | 0.94 (0.57, 1.55) | 1.15 (0.86, 1.55) | 1.03 (0.77, 1.37) |

AOR = adjusted odds ratio; CI = confidence interval; Covariates include Survey Wave, Age, Sex, Race/Ethnicity, Education Level, Cohabitation Status, Formal Sexuality Education Receipt.

[a]Reference group is 2011–2013.

[b]Reference group is the respondent did not discuss any topics.

[c]Statistically significant at $p < .05$.

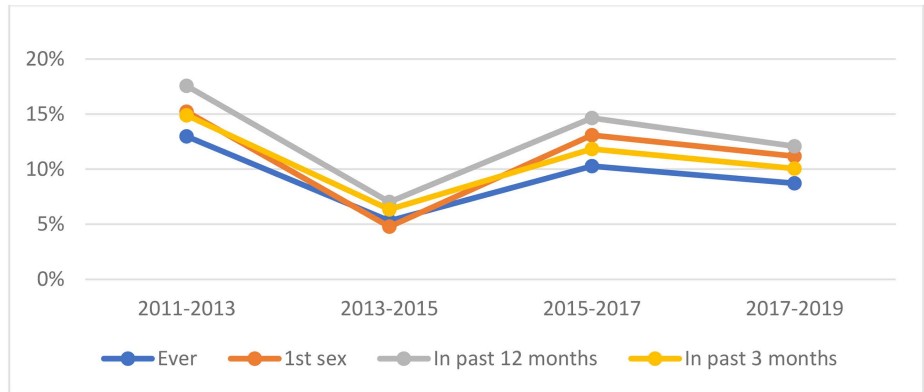

**Fig 2. Prevalence of withdrawal combined with at least one other contraceptive method across NSFG waves.**

## Limitations

This study is not devoid of limitations. First, this study does not explore how sexual orientation influences withdrawal use despite evidence that sexual orientation [51,52] and relationship status [53–56] are important factors in contraceptive choices. Second, withdrawal is often inconsistently measured and not traditionally considered a contraceptive method [39,53], potentially affecting respondents' perceptions of its validity as a contraception option. Third, self-reported data in the NSFG relies on respondents' recall of contraception use, which may introduce recall bias [57]. In combination with varying receipts of sexuality education, mastery of the content is not guaranteed, and further research should assess

**Table 4. Prevalence of withdrawal combined with at least one other contraceptive method in NSFG respondents aged 15-24 years between 2011 and 2019 (N = 14,262).**

| Predictor Variable | Ever | At first sex | In past 12 mos | In past 3 mos |
|---|---|---|---|---|
| | Weighted % (SE) | | | |
| Overall | 15.15 (0.58) | 5.89 (0.36) | 12.45 (0.52) | 10.50 (0.47) |
| Age (M, SE) | 20.61 (0.08) | 20.40 (0.11) | 20.58 (0.09) | 20.71 (0.09) |
| Sex | | | | |
| Female | 14.44 (0.75) | 4.27 (0.38) | 12.29 (0.69) | 10.59 (0.65) |
| Male | 15.84 (0.80) | 7.46 (0.59) | 12.61 (0.72) | 10.41 (0.61) |
| Race/Ethnicity | | | | |
| Hispanic | 14.80 (0.91) | 5.41 (0.57) | 12.14 (0.93) | 10.64 (0.93) |
| Non-Hispanic white | 15.57 (0.82) | 6.45 (0.56) | 12.64 (0.75) | 10.47 (0.65) |
| Non-Hispanic black | 15.35 (1.19) | 5.43 (0.64) | 13.03 (1.01) | 11.23 (1.04) |
| Non-Hispanic other | 12.91 (1.58) | 4.31 (0.86) | 10.88 (1.57) | 8.84 (1.64) |
| Education level | | | | |
| 9th grade or less | 5.59 (0.81) | 3.43 (0.67) | 4.05 (0.62) | 2.76 (0.49) |
| Some high school | 10.42 (0.84) | 3.93 (0.47) | 8.43 (0.80) | 6.83 (0.71) |
| GED/high school diploma | 17.34 (1.03) | 6.43 (0.65) | 14.12 (0.92) | 12.28 (0.85) |
| Some college | 20.39 (1.09) | 8.51 (0.85) | 17.25 (1.03) | 14.63 (0.99) |
| College degree | 19.32 (1.75) | 5.36 (0.86) | 16.04 (1.64) | 13.71 (1.48) |
| Graduate or professional degree | 30.22 (8.86) | 10.21 (4.41) | 23.46 (8.09) | 23.46 (8.09) |
| Has a religious affiliation | | | | |
| No | 16.18 (1.03) | 5.36 (0.57) | 13.50 (0.96) | 11.96 (0.92) |
| Yes | 14.76 (0.68) | 6.10 (0.43) | 12.04 (0.61) | 9.93 (0.49) |
| Currently cohabitating with a partner | | | | |
| No | 14.53 (0.59) | 5.79 (0.37) | 12.03 (0.53) | 9.85 (0.46) |
| Yes | 20.14 (1.71) | 6.67 (1.06) | 15.82 (1.45) | 15.72 (1.46) |
| Received any formal sexuality education | | | | |
| No | 11.20 (2.71) | 4.29 (1.34) | 7.96 (2.55) | 8.49 (2.69) |
| Yes | 15.27 (0.58) | 5.94 (0.37) | 12.58 (0.52) | 10.56 (0.47) |

respondents' proficiency in topics discussed with parents or caregivers. Finally, as a cross-sectional study, causal relationships between predictors and withdrawal use cannot be established.

Many variables impact one's contraception method decision including relationship power dynamics and status, women's concern of side effects, sexual partner knowledge, socioeconomic status, and insurance [58–60]. While these were not considered in this particular study, all of these are valuable factors to be considered in future research. This study expands on previous examinations of withdrawal as a contraceptive method in AYA samples by examining both withdrawal as the only method and withdrawal combined with at least one other method. Further, the current study is the first to specifically examine how receipt of informal sexuality education is associated with withdrawal, providing additional information about how certain informal sexuality topics are associated with withdrawal use at multiple recall and reference periods instead of only examining withdrawal at last sex or first sex.

The results of this study provide greater impetus for providers of informal sexuality education (i.e., peers, parents, siblings) to have additional resources and support made available to them. Programs and interventions targeting this mode of delivery should have access to evidence-based materials which could then be used to deliver informal sexuality education topics. Doing so would complement any additional formal sexuality education and present another avenue for

**Table 5. Binary logistic regression results of the association between topics of informal sexuality education and using withdrawal combined with at least one other contraceptive method among adolescents and young adults aged 15-24 years in NSFG between 2011 and 2019 (N = 14,262).**

| Predictor Variable | Ever | At first sex | In past 12 mos | In past 3 mos |
|---|---|---|---|---|
| Survey Wave[a] | AOR (95% CI) | AOR (95% CI) | AOR (95% CI) | AOR (95% CI) |
| 2013-2015 | 1.05 (0.76, 1.44) | 0.76 (0.48, 1.21) | 1.21 (0.86, 1.68) | 1.28 (0.91, 1.80) |
| 2015-2017 | 1.28 (0.95, 1.73) | 1.14 (0.76, 1.73) | **1.42 (1.04, 1.95)**[c] | **1.45 (1.05, 2.01)**[c] |
| 2017-2019 | 1.05 (0.79, 1.40) | 1.06 (0.71, 1.58) | 1.10 (0.81, 1.49) | 1.16 (0.83, 1.61) |
| Sexuality education topic discussed[b] | | | | |
| Any topics | 0.91 (0.67, 1.25) | 0.80 (0.49, 1.30) | 1.04 (0.75, 1.44) | 1.06 (0.74, 1.51) |
| Sexually transmitted infections | **0.80 (0.67, 0.96)**[c] | 1.08 (0.84, 1.40) | **0.79 (0.65, 0.96)**[c] | 0.85 (0.68, 1.06) |
| How to prevent HIV/AIDS | 0.86 (0.71, 1.05) | 0.99 (0.72, 1.38) | 0.87 (0.71, 1.08) | 0.78 (0.61, 1.01) |
| How to say no to sex | 1.17 (0.93, 1.47) | 0.91 (0.59, 1.39) | 1.12 (0.89, 1.42) | 1.27 (0.99, 1.64) |
| How to wait until marriage to have sex | 1.18 (0.94, 1.49) | 0.99 (0.71, 1.37) | 1.20 (0.94, 1.53) | 1.24 (0.95, 1.61) |
| How to use a condom | 0.92 (0.72, 1.16) | 1.01 (0.73, 1.38) | 0.88 (0.69, 1.12) | 0.81 (0.63, 1.04) |
| Methods of birth control | **1.33 (1.09, 1.63)**[c] | 1.34 (0.99, 1.81) | **1.38 (1.11, 1.71)**[c] | **1.30 (1.01, 1.67)**[c] |
| Where to get birth control | 1.20 (0.95, 1.51) | 1.31 (0.95, 1.82) | 1.13 (0.87, 1.46) | 1.02 (0.77, 1.35) |

AOR = adjusted odds ratio; CI = confidence interval; Covariates include Survey Wave, Age, Sex, Race/Ethnicity, Education Level, Cohabitation Status, Formal Sexuality Education Receipt.

[a]Reference group is 2011–2013.

[b]Reference group is the respondent did not discuss any topics.

[c]Statistically significant at $p < .05$.

reducing negative sexual health outcomes. While reported receipt of formal sexuality education is decreasing, rates of withdrawal as a contraceptive method are *increasing*, and, given the most recent shift toward home-based learning (i.e., virtual classrooms due to the COVID-19 pandemic), it is critical now more than ever for AYAs to receive evidence-based sexuality education wherever they are. This makes clear the need for a better understanding of how informal sexuality education is associated with sexual behaviors (i.e., contraceptive use) and how to implement and evaluate interventions targeting informal sources.

## Author contributions

**Conceptualization:** John L. Ferrand, Arthur H. Owora, Eric R. Walsh-Buhi.

**Data curation:** John L. Ferrand, Arthur H. Owora.

**Formal analysis:** John L. Ferrand, Arthur H. Owora.

**Methodology:** John L. Ferrand, Arthur H. Owora, Eric R. Walsh-Buhi.

**Supervision:** Eric R. Walsh-Buhi.

**Writing – original draft:** John L. Ferrand, Arthur H. Owora, Eric R. Walsh-Buhi.

**Writing – review & editing:** John L. Ferrand, Arthur H. Owora, Alexandra T. Hughes-Wegner, Eric R. Walsh-Buhi.

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
