## [Decision Letter · Decision Letter 0]

19 Aug 2024

Dear Dr. Walsh-Buhi,

Thank you for submitting your manuscript to PLOS ONE. After careful consideration, we feel that it has merit but does not fully meet PLOS ONE’s publication criteria as it currently stands. Therefore, we invite you to submit a revised version of the manuscript that addresses the points raised during the review process.

We look forward to receiving your revised manuscript.

Kind regards,

Malith Kumarasinghe, MBBS MD

Academic Editor

PLOS ONE

Additional Editor Comments:

Overall

Please define the age group of youth as there are few differences in definitions globally.

Does the national survey capture the socioeconomic status in the national survey? This could be a valuable covariate!

It is better to include overall and modern contraceptive prevalence among youth along with the results of tables 2 & 4 to have an overall idea!

Lines 40-42-Abstract; Conclusion- Please reword

Line-67- reference at completion of the sentence!

Lines 77-81- Please reword the sentence!

Lines 87-89- Purpose does not match with the title and the objective given in the abstract! Please consider using “youth”.

Line 102- Please explain the reason for exclusion of pregnant youth. Specially with questions like “define withdrawal use: 1) ever use, 2) at first sex, 3) past 12 months, and 4) 109 past 3 months.[24]”

Results

Table 1 one- Will it possible to compare the demographic with national US data?

Table 1- Education level could be better presented with breakdown of the youth group say 15-19 and above 19 which could reduce the misinterpretation of higher representation of 9th grade or less/ high school in study cohort.

Table 2 & 4- It is better if you could present the survey wave results (overall) for the four variables in graph format for better presentation! The 1st 4 rows of the table 2.

Lines 190-192- Discussion- It is important that you explore the reasons for comparatively higher prevalence in 3rd wave and subsequent drop in 4th wave. This is an important discussion point!

Lines199-205- What is the influence of social media/ online content in determining the contraceptive practice!

Reviewers' comments:

Reviewer's Responses to Questions

**Comments to the Author**

1. Is the manuscript technically sound, and do the data support the conclusions?

Reviewer #1: Yes

Reviewer #2: Yes

2. Has the statistical analysis been performed appropriately and rigorously?

Reviewer #1: Yes

Reviewer #2: Yes

3. Have the authors made all data underlying the findings in their manuscript fully available?

Reviewer #1: Yes

Reviewer #2: Yes

4. Is the manuscript presented in an intelligible fashion and written in standard English?

Reviewer #1: Yes

Reviewer #2: Yes

Reviewer #1: May 20, 2024

Review of manuscript PONE-D-24-14716 entitled "Prevalence and correlates of withdrawal as contraceptive method in a nationally representative sample of youth: A cross-sectional study". The manuscript examines the trend of withdrawal use (exclusively or in combination with other methods) and its relation with the receipt of sexual education. The results contribute to the literature. The following issues need to be addressed before it can be publishable.

- P 4, LL 66-67; 73-74: The authors discuss that withdrawal is less effective method, while the failure rate of "correct" and "consistent" use of withdrawal is 4%. The higher failure rate of withdrawal is referred to its "typical use" that is estimated to be 20% in the USA and more or less in other parts of world (suggest reading this paper https://doi.org/10.1038%2Fs41598-023-37398-1). The authors need to specify that they are talking about the failure rate of the typical use of withdrawal. Studies on withdrawal usually tend to undermine the importance of this method for women's health (no side effects) and its wide availability, and its role in fertility transition.

- P. 5, LL83-84: The literature shows that the internet is the main source of sexuality education rather parents and siblings.

- Introduction: the importance of studying the relationship between sexuality education and withdrawal use has not been well stated. As mentioned, withdrawal is not part of family planning and sexual formal education of health systems. So, studying its relationship with withdrawal use does not sound clear to me here. The review of literature on this topic is very limited and short in this manuscript.

- L. 105: "Primary Measures" is an unusual term used for "dependent variables). Please consider revising it to "outcome" or "dependent" variables.

- There is a lack of theoretical logic for selecting covariates. Except the measures of sexuality education, al demographic variables have been studied by past studies. They are not the determinants of withdrawal use. I would suggest reviewing a tick literature on the relationship between sexuality education, as the source of contraceptive knowledge, and contraceptive use, which makes a good background for studying this relationship.

- L. 135: Please specify the type of logistic regression used and the justification for its use.

- The findings in LL. 171-174 is one of the key results of this study that needs an explanation in Discussion section. Please make sure to expand the discussion of why the receipt of informal sexuality education related to other methods of birth control was associated with the greater risk of using withdrawal in LL 198-202. Can this be due to the fact that those receiving education on the side effects of modern birth control (especially pills) are more likely to use withdrawal with combination of condoms in order to avoid the side effects of Pills and other hormonal methods?

- L 194: Revise " One potential reason for reason for".

Reviewer #2: Overall comment:

Thank you for the opportunity to review this manuscript. I appreciate the authors' efforts in conducting this study which explores an important area.The manuscript shows promise but requires further development. The research question is interesting, but the discussion could be strengthened. Minor revisions are recommended. Please see specific comments below.

Abstract

Results:

•The author mentions the prevalence of withdrawal is higher in combination than alone (lines 34-35). I would suggest adding “across all waves and reference period” and accompanying it with values to improve clarity.

•Suggest to report values for outputs from the regression model; AOR with CI

Conclusion:

broad and ambiguous. It could be precise and specific.

Main Manuscript

Methods:

Line 127- formal sexual education: it would be useful to mention the various sources similar to the types of informal sources mentioned.

Results:

•Line 166 refers to Table 3, it should be Table 4

•Table 4: Prevalence of non-Hispanic other population under ever use is reported incorrectly

Discussion:

•The findings of this study are based on contraceptive use that dates back to 5 years ago. The pattern of contraceptive use likely changed since the data collection for this study. If the finding were to be relevant even today to draw meaningful conclusions based on which policy decisions are made, it would be useful to compare patterns of contraceptive use from recent literature in the absence of updated prevalence rates of withdrawal in particular. This could strengthen the findings of the current study although not updated.

•A great many important covariates that have a direct influence on contraceptive use have not been explored in the study which is a major limitation and requires mentioning. Ex: previous experience, marital status, women's concerns on side effects preventing them from accessing other birth control methods, male partners' influence on deciding, power dynamics within relationships, educational level of the partner, cultural norms, economic status and cost of health insurance cover, sexual behaviour etc. However, given that the study was based on secondary data which would have failed to capture crucial factors, it would be useful to discuss the potential influence of these factors on the findings.

•Prevalence alone or in combination with other methods increased from 2011 to 2017 and then declined across all reference periods except for the 2013-2015 wave at first sex (Table 2). This trend observed needs an explanation.

•Line 199-202: The authors' explanation for higher odds of withdrawal used in combination with other methods is ambiguous. A possible explanation would be a sense of safety perceived when combined with other more effective methods. It still fails to explain the continuation of withdrawal despite being better informed.

•Line 211 needs a reference for this claim.

**Do you want your identity to be public for this peer review?** For information about this choice, including consent withdrawal, please see our Privacy Policy

Reviewer #1: No

Reviewer #2: No

---

## [Author Response · Author response to Decision Letter 1]

11 Dec 2024

Editor comments

Please define the age group of youth as there are few differences in definitions globally.

Author response: We are delighted for the opportunity to revise and resubmit our manuscript to PLOS ONE. We have addressed each of the Reviewers’ comments by either making a change/edit in the revised manuscript or providing further rationale for our original decision (outlined in detail below). As a result of the Reviewer comments, we believe the revised manuscript has been greatly strengthened and we hope that you agree. Please do not hesitate to let us know if you have any questions or if you require additional changes/edits. Thank you in advance. We look forward to hearing back from you!

Regarding age definitions of youth: Thank you for pointing this out! This comment has been addressed, information has been added to the introduction and methods section.

Does the national survey capture the socioeconomic status in the national survey? This could be a valuable covariate!

Author response: This is an excellent question. While this may be a valuable covariate, this was not the focus of our present study. Based on a different reviewer comment, however, additional information was added to the limitations section regarding what covariates were included or excluded.

Lines 40-42-Abstract; Conclusion- Please reword

Author response: This comment has been addressed.

Line-67- reference at completion of the sentence!

Author response: This comment has been addressed.

Lines 77-81- Please reword the sentence!

Author response: This comment has been addressed.

Lines 87-89- Purpose does not match with the title and the objective given in the abstract! Please consider using “youth”.

Author response: This comment has been addressed.

Line 102- Please explain the reason for exclusion of pregnant youth. Specially with questions like “define withdrawal use: 1) ever use, 2) at first sex, 3) past 12 months, and 4) 109 past 3 months.[24]”

Author response: Thank you for this comment. We decided to exclude pregnant youth to get an unbiased sample of those using a form of contraception regularly and without having been pregnant before.

Table 1 one- Will it possible to compare the demographic with national US data? The NSFG collects data using computer-assisted personal interviews of men and women aged 15-49 years in the United States. The NSFG utilizes a stratified, multistage, clustered, probability sampling design to report nationally representative estimates, meaning that the survey is designed to be nationally representative of men and women aged 15-49 living in households in the United States. As these data are already nationally representative, we feel there is no added value to report duplicate demographic characteristics in the table.

Table 1- Education level could be better presented with breakdown of the youth group say 15-19 and above 19 which could reduce the misinterpretation of higher representation of 9th grade or less/ high school in study cohort.

Author response: Thank you for this suggestion. We feel that would misrepresent the data since the item from the NSFG asks about the highest education level respondents reported receiving in their life (not what level of education they are currently receiving or are currently in).

Table 2 & 4- It is better if you could present the survey wave results (overall) for the four variables in graph format for better presentation! The 1st 4 rows of the table 2.

Author response: Thank you! We have removed the wave prevalence rows in tables 2 and 4 and have added two graphs depicting these prevalences over time for each outcome at each timepoint. Please see the revised Figures 1 and 2.

Lines 190-192- Discussion- It is important that you explore the reasons for comparatively higher prevalence in 3rd wave and subsequent drop in 4th wave. This is an important discussion point!

Author response: After further review, we agree. We have added additional information to the discussion section to address this comment (please see lines 222-225).

Lines199-205- What is the influence of social media/ online content in determining the contraceptive practice!

Author response: Additional information has been added throughout the manuscript.

Reviewer 1

Review of manuscript PONE-D-24-14716 entitled "Prevalence and correlates of withdrawal as contraceptive method in a nationally representative sample of youth: A cross-sectional study". The manuscript examines the trend of withdrawal use (exclusively or in combination with other methods) and its relation with the receipt of sexual education. The results contribute to the literature. The following issues need to be addressed before it can be publishable.

Author response: Thank you for taking time to review our manuscript. Your feedback has strengthened our manuscript, and we are eager for the follow-up review.

- P 4, LL 66-67; 73-74: The authors discuss that withdrawal is less effective method, while the failure rate of "correct" and "consistent" use of withdrawal is 4%. The higher failure rate of withdrawal is referred to its "typical use" that is estimated to be 20% in the USA and more or less in other parts of world (suggest reading this paper https://doi.org/10.1038%2Fs41598-023-37398-1). The authors need to specify that they are talking about the failure rate of the typical use of withdrawal. Studies on withdrawal usually tend to undermine the importance of this method for women's health (no side effects) and its wide availability, and its role in fertility transition.

Author response: This comment had been addressed and additional information has been added.

- P. 5, LL83-84: The literature shows that the internet is the main source of sexuality education rather parents and siblings.

This comment has been addressed.

- Introduction: the importance of studying the relationship between sexuality education and withdrawal use has not been well stated. As mentioned, withdrawal is not part of family planning and sexual formal education of health systems. So, studying its relationship with withdrawal use does not sound clear to me here. The review of literature on this topic is very limited and short in this manuscript.

Author response: This comment has been addressed.

- L. 105: "Primary Measures" is an unusual term used for "dependent variables). Please consider revising it to "outcome" or "dependent" variables.

Author response: This comment has been addressed.

- There is a lack of theoretical logic for selecting covariates. Except the measures of sexuality education, al demographic variables have been studied by past studies. They are not the determinants of withdrawal use. I would suggest reviewing a tick literature on the relationship between sexuality education, as the source of contraceptive knowledge, and contraceptive use, which makes a good background for studying this relationship.

Author response: This comment has been addressed.

- L. 135: Please specify the type of logistic regression used and the justification for its use.

Author response: Thank you for the request for clarification! We have added more detail in lines 165-168 of the revised manuscript to address this comment.

- The findings in LL. 171-174 is one of the key results of this study that needs an explanation in Discussion section. Please make sure to expand the discussion of why the receipt of informal sexuality education related to other methods of birth control was associated with the greater risk of using withdrawal in LL

198-202. Can this be due to the fact that those receiving education on the side effects of modern birth control (especially pills) are more likely to use withdrawal with combination of condoms in order to avoid the side effects of Pills and other hormonal methods?

Author response: This comment has been addressed.

- L 194: Revise " One potential reason for reason for".

Author response: This comment has been addressed.

Reviewer #2

Thank you for the opportunity to review this manuscript. I appreciate the authors' efforts in conducting this study which explores an important area. The manuscript shows promise but requires further development. The research question is interesting, but the discussion could be strengthened. Minor revisions are recommended. Please see specific comments below.

Author response: Thank you for taking the time to review the manuscript. We appreciate your thorough review and believe the addressed edits have strengthened the manuscript.

Abstract

Results:

•The author mentions the prevalence of withdrawal is higher in combination than alone (lines 34-35). I would suggest adding “across all waves and reference period” and accompanying it with values to improve clarity.

•Suggest to report values for outputs from the regression model; AOR with CI

Conclusion:

broad and ambiguous. It could be precise and specific.

We thank the reviewer for these comments. We have added the suggested blurb as well as the statistical results into the revised abstract. Please see the revised manuscript with track changes (lines 34-38).

Main Manuscript

Methods:

Line 127- formal sexual education: it would be useful to mention the various sources similar to the types of informal sources mentioned.

This comment has been addressed throughout the manuscript.

Results:

•Line 166 refers to Table 3, it should be Table 4

This comment has been addressed.

•Table 4: Prevalence of non-Hispanic other population under ever use is reported incorrectly

Author response: Thank you for catching this error. We have corrected it in the table.

Discussion:

•The findings of this study are based on contraceptive use that dates back to 5 years ago. The pattern of contraceptive use likely changed since the data collection for this study. If the finding were to be relevant even today to draw meaningful conclusions based on which policy decisions are made, it would be useful to compare patterns of contraceptive use from recent literature in the absence of updated prevalence rates of withdrawal in particular. This could strengthen the findings of the current study although not updated.

Author response: The most recent NSFG data is the 2017-2019 cohort. However, recent literature has been added throughout the manuscript to strengthen our findings.

•A great many important covariates that have a direct influence on contraceptive use have not been explored in the study which is a major limitation and requires mentioning. Ex: previous experience, marital status, women's concerns on side effects preventing them from accessing other birth control methods, male partners' influence on deciding, power dynamics within relationships, educational level of the partner, cultural norms, economic status and cost of health insurance cover, sexual behaviour etc. However, given that the study was based on secondary data which would have failed to capture crucial factors, it would be useful to discuss the potential influence of these factors on the findings.

Author response: Additional information has been added to our limitations section.

•Prevalence alone or in combination with other methods increased from 2011 to 2017 and then declined across all reference periods except for the 2013-2015 wave at first sex (Table 2). This trend observed needs an explanation.

Author response: We have added more information to try to explain this trend.

•Line 199-202: The authors' explanation for higher odds of withdrawal used in combination with other methods is ambiguous. A possible explanation would be a sense of safety perceived when combined with other more effective methods. It still fails to explain the continuation of withdrawal despite being better informed.

Author response: This comment has been addressed, and additional information and insights have been added throughout the discussion section.

•Line 211 needs a reference for this claim.

Author response: This comment has been addressed.

---

## [Decision Letter · Decision Letter 1]

17 Mar 2025

Dear Dr. Walsh-Buhi,

Thank you for submitting your manuscript to PLOS ONE. After careful consideration, we feel that it has merit but does not fully meet PLOS ONE’s publication criteria as it currently stands. Therefore, we invite you to submit a revised version of the manuscript that addresses the points raised during the review process.

We look forward to receiving your revised manuscript.

Kind regards,

Malith Kumarasinghe, MBBS MD

Academic Editor

PLOS ONE

Journal Requirements:

Additional Editor Comments:

Dear Authors,

Please carefully address the comments and concerns of the reviewers.

Thank you

Reviewers' comments:

Reviewer's Responses to Questions

**Comments to the Author**

Reviewer #2: (No Response)

Reviewer #3: (No Response)

Reviewer #4: (No Response)

2. Is the manuscript technically sound, and do the data support the conclusions?

Reviewer #2: Yes

Reviewer #3: Yes

Reviewer #4: Yes

3. Has the statistical analysis been performed appropriately and rigorously?

Reviewer #2: Yes

Reviewer #3: Yes

Reviewer #4: Yes

4. Have the authors made all data underlying the findings in their manuscript fully available?

Reviewer #2: Yes

Reviewer #3: Yes

Reviewer #4: Yes

5. Is the manuscript presented in an intelligible fashion and written in standard English?

Reviewer #2: Yes

Reviewer #3: Yes

Reviewer #4: Yes

Reviewer #2: The authors have addressed most of the comments adequately. However, the following still needs to be addressed:

1.Abstract- results- although prevalence has been reported, it is best reported accompanied with Confidence intervals. (Interval estimates).

2.The original comment “Methods: Line 127- formal sexual education: it would be useful to mention the various sources similar to the types of informal sources mentioned”. - not addressed, although the authors say they have! Line 157-159 in the revised manuscript.

Reviewer #3: Reviewer comments

Thank you for the opportunity to review this revised manuscript – PONE-D-24-14716_R1. Most of the editor comments and two reviewers’ comments were addressed. Recommend to publish with the following minor revisions.

1. Defining the age group of youth should be clearer and should be unique throughout the manuscript.

2. In the line 66 – put the reference at the completion of the sentence.

3. Lines 88 – 90 consider using “youth” still not addressed properly.

Reviewer #4: This was an interesting article researching a currently relevant topic. Congratulations on a job well done!

I have three concerns, as all others have been addressed by previous reviewers.

My first concern is in the discussion section line 213-216, where you describe "the increase of withdrawal use during the 2015-2017 wave may be partially explained by the beginning of the Donald Trump administration which heavily defunded comprehensive sexual health programs between 2016 and 2020".

- the timing for the explanation does not line up as the change happens before the administration comes in to power. Therefore, your concusion loses it's validity.

My second cencern is the clain that this is an "update". line 195-197. "The goal of the current study was to provide an update of the prevalence of withdrawal as a contraceptive method in a nationally representative sample of American youth as well as to examine the relationship between informal sexuality education and withdrawal".

- as the most recent data is from 2019, you may need to clarify regarding the availability of the data. That is, the most recent available NSFG data is from ........

My third concern is that the persons delivering formal sexual education have not been specified or identified. That is, the receipt/ acceptance of sexual education may differ on the expertise/ perceived reliability of the individual delivering it. Thus it would be great to have it addressed in the writing.

Thank you.

**Do you want your identity to be public for this peer review?** For information about this choice, including consent withdrawal, please see our Privacy Policy

Reviewer #2: No

Reviewer #3: No

Reviewer #4: No

---

## [Author Response · Author response to Decision Letter 2]

17 Jul 2025

Reviewer #2: The authors have addressed most of the comments adequately. However, the following still needs to be addressed:

Abstract- results- although prevalence has been reported, it is best reported accompanied with Confidence intervals. (Interval estimates).

AUTHOR RESPONSE: As this was not a statistical comparison, we have added the individual standard errors to accompany the prevalences. This comment has been addressed.

The original comment “Methods: Line 127- formal sexual education: it would be useful to mention the various sources similar to the types of informal sources mentioned”. - not addressed, although the authors say they have! Line 157-159 in the revised manuscript.

AUTHOR RESPONSE: Thank you for taking time to review this document. Additional information has been added to the introduction (lines 91-94) and methods section (line 146-147).

Reviewer #3: Reviewer comments

Thank you for the opportunity to review this revised manuscript – PONE-D-24-14716_R1. Most of the editor comments and two reviewers’ comments were addressed. Recommend to publish with the following minor revisions.

AUTHOR RESPONSE: Thank you for your careful consideration of our document. Below, the following comments have been addressed and can be found at the associated line numbers in red.

1. Defining the age group of youth should be clearer and should be unique throughout the manuscript.

AUTHOR RESPONSE: In response to the reviewer comment, we have deleted our use of the term “Youth”, as definitions vary across the globe, and replaced it with a more current and accurate reference to young people this age, “Adolescents and young adults” or AYAs. This change has been made throughout the manuscript, including in the title, abstract, and table headers.

2. In the line 66 – put the reference at the completion of the sentence.

AUTHOR RESPONSE: This comment has been addressed. We have also “cleaned up” the citations throughout the revised manuscript, ensuring consistency and adherence to journal guidelines.

3. Lines 88 – 90 consider using “youth” still not addressed properly.

AUTHOR RESPONSE: Thank you for drawing our attention to this oversight. As noted above, we have deleted our use of the term “Youth” and replaced it with a more current and accurate reference to young people this age, “Adolescents and young adults” or AYAs. This change has been made throughout the manuscript, including in the title, abstract, and table headers.

Reviewer #4: This was an interesting article researching a currently relevant topic. Congratulations on a job well done! I have three concerns, as all others have been addressed by previous reviewers.

My first concern is in the discussion section line 213-216, where you describe "the increase of withdrawal use during the 2015-2017 wave may be partially explained by the beginning of the Donald Trump administration which heavily defunded comprehensive sexual health programs between 2016 and 2020".

- the timing for the explanation does not line up as the change happens before the administration comes into power. Therefore, your conclusion loses it's validity.

AUTHOR RESPONSE: Thank you for drawing our attention to this! This comment has been addressed (line 214-221).

My second cencern is the clain that this is an "update". line 195-197. "The goal of the current study was to provide an update of the prevalence of withdrawal as a contraceptive method in a nationally representative sample of American youth as well as to examine the relationship between informal sexuality education and withdrawal".

- as the most recent data is from 2019, you may need to clarify regarding the availability of the data. That is, the most recent available NSFG data is from ........

AUTHOR RESPONSE: The comment has been addressed.

My third concern is that the persons delivering formal sexual education have not been specified or identified. That is, the receipt/ acceptance of sexual education may differ on the expertise/ perceived reliability of the individual delivering it. Thus it would be great to have it addressed in the writing.

AUTHOR RESPONSE: This comment has been addressed. Based on a previous reviewer comment, additional information has been added to the introduction and methods section to give a clear definition of where formal sexual education may have been received (see lines 91-94, as an example).

---

## [Decision Letter · Decision Letter 2]

18 Dec 2025

The Role of Sex Education in Withdrawal Use: Prevalence and Correlates Among a Nationally Representative Sample of Adolescents and Young Adults

PONE-D-24-14716R2

Dear Dr. Walsh-Buhi,

We’re pleased to inform you that your manuscript has been judged scientifically suitable for publication and will be formally accepted for publication once it meets all outstanding technical requirements.

Kind regards,

Jianhong Zhou

Staff Editor

PLOS One

Additional Editor Comments (optional):

Reviewers' comments:

Reviewer's Responses to Questions

**Comments to the Author**

Reviewer #2: All comments have been addressed

2. Is the manuscript technically sound, and do the data support the conclusions?

Reviewer #2: Yes

3. Has the statistical analysis been performed appropriately and rigorously?

Reviewer #2: Yes

4. Have the authors made all data underlying the findings in their manuscript fully available?

Reviewer #2: Yes

5. Is the manuscript presented in an intelligible fashion and written in standard English?

Reviewer #2: Yes

Reviewer #2: I am happy to note that the authors have attempted to address all concerns. However, I would like to stress that while standard error can be used to describe the uncertainty in a prevalence estimate, confidence intervals are generally preferred because they provide a more informative and easily interpretable measure of the precision of the estimate.

In light of lack of data to compute confidence intervals I suppose it is acceptable to use standard error values.

I recommned the acceptance of the manuscript in its current form.

**Do you want your identity to be public for this peer review?** For information about this choice, including consent withdrawal, please see our Privacy Policy

Reviewer #2: No

---

## [Editor Report · Acceptance letter]

PONE-D-24-14716R2

PLOS One

Dear Dr. Walsh-Buhi,

I'm pleased to inform you that your manuscript has been deemed suitable for publication in PLOS One. Congratulations! Your manuscript is now being handed over to our production team.

Kind regards,

on behalf of

Dr. Jianhong Zhou

Staff Editor

PLOS One